



# Globally- and Hemispherically-Integrated Joule heating rates during the 17 March 2015 geomagnetic storm, according to Physics-based and Empirical Models

Stelios Tourgaidis[1], Dimitris Baloukidis[1], Panagiotis Pirnaris[1], Theodoros Sarris[1], Aaron Ridley[2], and Gang Lu[3]

[1]Department of Electrical and Computer Engineering, Democritus University of Thrace,Xanthi, Greece
[2]Department of Atmosphere, Oceanic and Space Sciences, University of Michigan,Michigan,USA
[3]High Altitude Observatory, National Center for Atmospheric Research,Boulder, Colorado, USA

**Correspondence:** Stelios Tourgaidis (stourgai@ee.duth.gr), Theodoros Sarris (tsarris@ee.duth.gr)

**Abstract.** It is well known that the primary solar wind energy dissipation mechanism in the Earth's upper atmosphere is Joule heating. Two of the most commonly used physics-based Global Circulation Models (GCM) of the Earth's upper atmosphere are the Global Ionosphere/ Thermosphere Model (GITM) and the Thermosphere-Ionosphere-Electrodynamics General Circulation Model (TIEGCM). Both are externally driven by models that provide the specification of high-latitude electric fields as well as auroral precipitation. In this study, a comparison of the evolution of the globally-integrated Joule heating rates is performed be-
tween the two physics-based models, TIEGCM and GITM, each driven by two different specifications of high-latitude electric fields, namely the Weimer 2005 and the Assimilative Mapping of Ionospheric Electrodynamics (AMIE) models. Several empirical formulations provide estimates of Joule heating based on solar and geomagnetic activity indices; a further comparison is performed between these empirical formulations and the GCMs. It is found that all empirical formulations underestimate
Joule heating rates compared to GITM and TIEGCM, whereas TIEGCM calculates lower heating rates compared to GITM, both when Weimer 2005 and AMIE models are used as drivers. By calculating the heating rates in the northern and southern hemispheres it is found that in GITM and TIEGCM higher Joule heating rates are observed in the southern hemisphere, when the Weimer model is used. These discrepancies disappear when the AMIE method is used. In that case higher Joule heating rates are calculated for the northern hemisphere. The differences and similarities between the two GCMs and the empirical
models are discussed.

## 1 Introduction

During geomagnetic storms, Joule heating is known to be the dominant solar wind energy dissipation mechanism. Joule heating maximizes in the lower thermosphere-ionosphere (LTI) region, within the 100 to 200 km altitude range, where current density and conductivity (Pedersen and Hall) maximize Huang et al. (2012); Baloukidis et al. (2023). The quantification of Joule
heating is a subject of intense research, as it is critical in determining the structure and evolution of the Lower Thermosphere-Ionosphere, and is responsible for a number of effects of societal importance, such as for determining atmospheric drag and



predicting the resulting deorbiting times of satellites and space debris within this region Palmroth et al. (2021); Sarris et al. (2023a). For example, the loss of 40 Space-X satellites in February 2022 is thought to have been caused by an underestimate of the enhancement of thermospheric neutral density that resulted from enhanced Joule heating during a moderate geomagnetic

storm Dang et al. (2022); Zhang et al. (2022b); Hapgood et al. (2022). It is for this reason that quantifying the heating rates is critical in order to accurately determine satellite drag and orbital lifetime estimations.

Whereas the physics of the collisional processes leading to Joule heating is well understood and is captured in Global Circulation Models (GCMs) of the ionosphere-thermosphere (IT) system, the quantification of Joule heating is still largely unknown, and large discrepancies appear between different models and estimation methodologies Palmroth et al. (2005);

Rodger et al. (2001). This is in part because the exact quantification of Joule heating requires the simultaneous and co-located measurement of all relevant parameters that are involved in the calculations of conductivity, electrical currents and fields, and in part because an unknown amount of Joule heating is found in small-scale or sub-grid variability that can not be captured by current models. Also contributing to the above uncertainty, the lower thermosphere-ionosphere (LTI) region, where Joule heating maximizes, is the least sampled of all atmospheric regions (see, e.g., Sarris et al., 2023a) and references therein): due to

the large air drag, the altitude range from ∼100 to 200 km is too high for balloon experiments and too low for current Low-Earth Orbit (LEO) satellites. Thus, the majority of available measurements for this region comes from ground based observatories, such as Incoherent Scatter Radars, and very few in-situ space missions, such as the Atmosphere Explorers of the early 80'. Measurements from the above are used in formulating empirical models of the upper atmosphere, such as the International Reference Ionosphere (IRI) Bilitza (2018), NRLMSISE-00 Picone et al. (2002) and the Horizontal Wind Model (HWM) Drob

et al. (2008). Furthermore, physics-based global circulation Models (GCM), such as the Global Ionosphere/Thermosphere Model (GITM) Ridley et al. (2006) or the National Center for Atmospheric Research (NCAR) Thermosphere-Ionosphere-Electrodynamics General Circulation Model (TIEGCM) Qian et al. (2014) simulate the energetics, dynamics and chemistry of this region. However, there are great discrepancies in describing the basic state of the LTI between empirical models and physics-based models, such as neutral temperature and density, which is largely due to the uncertainty in estimating the amount

of Joule heating in the LTI.

Among physics-based models, GITM and TIEGCM are widely used by the upper atmosphere scientific community. Both are 3D gridded numerical models that are used to simulate the state of the thermosphere and ionosphere in response to external driving by solar wind conditions. GITM and TIEGCM are both based on a set of equations that describe the physical processes that occur within the thermosphere and ionosphere, such as radiation, convection, and dynamical forcing. From the outputs

of these models, which include all essential variables or geophysical observables of the thermosphere and ionosphere, Joule heating can be directly computed at each model grid point.

Together with the above physics-based models, a number of empirical formulations have been derived as proxies of Joule heating, driven by solar and geomagnetic conditions. For example, Joule heating has been found to be closely related to the AE and AL indices (see, e.g., Perreault and Akasofu, 1978; Akasofu, 1981; Ahn et al., 1983; Baumjohann and Kamide, 1984;

Ahn et al., 1989; Richmond et al., 1990; Cooper et al., 1995; Lu et al., 1995, 1998). Seasonal and hemispherical differences have been examined as well to establish a more accurate relation between Joule heating and the geomagnetic indices Nisbet





(1982); Lu et al. (1998). Further to the above, Chun et al. (1999) estimated Joule heating with a quadratic fit to the Polar Cap ($PC$) index, whereas Knipp et al. (2005) expanded on the work of Chun et al. (1999) by proposing a formula that is based on both the PC and the Disturbance Storm Time ($Dst$) indices; and Weimer (2005) calculated Joule heating empirically, based on

a model of Poynting flux that is derived using measurements of the Dynamics Explorer 2 satellite. It is noted that most of the above empirical formulations do not take into account the effects of neutral winds, which are known to impact Joule heating significantly (see, e.g., Lu et al., 1995; Emery et al., 1999).

     Empirical models are often designed to describe large-scale climatology and can thus underestimate the real-time magnetospheric energy input. Such differences between empirical models and observations have been discussed extensively in the

literature (see, e.g., Cosgrove and Codrescu, 2009; Cosgrove et al., 2011). Various data assimilation methods have been developed to mitigate the discrepancy, e.g., the standard AMIE procedure Richmond and Kamide (1988a) and methods like SECS (Amm, 1997), LDFF (Bristow et al., 2016) and Lattice Kriging (Wu and Lu, 2022). Data assimilation methods are used to replace the empirical high-latitude drivers in GCMs, resulting in higher levels of variability, and enhanced Joule heating. These usually show a better agreement with observations. Some examples of data-driven modeling include the studies by Lu et al.

(2020), who performed extensive comparisons between simulation results from TIEGCM using AMIE as a driver, and Lu et al. (2023), who also used TIEGCM using the Lattice Kriging method to derive their high-latitude auroral and convection patterns.

     In the following, Joule heating estimates are calculated and presented based on simulation results of the solar storm of 17 March 2015, the largest geomagnetic storm of solar cycle 24, which is also known as St Patrick's day 2015 storm. As part of this study, globally integrated Joule heating rates are calculated in both GITM and TIEGCM, and are compared against estimates

obtained from various empirical formulations. GITM and TIEGCM simulations are performed using both the Weimer 2005 empirical high-latitude electric field model (Weimer, 2005) and the AMIE data assimilation method Richmond and Kamide (1988a) as drivers. It is noted that the integration of other available electric field data assimilation models such as those listed above (i.e., the models by Richmond and Kamide (1988b), Amm (1997), Bristow et al. (2016) and Wu and Lu (2022)) with TIEGCM and GITM was not available to the authors at the time of this study, and is not considered herein. The integration of

the above data assimilation methods as drivers of commonly used GCMs is a topic of future research. Joule heating estimates are presented as time series over the course of St Patrick's day 2015 storm. Together with the time series of the evolution of Joule heating during the storm, the cumulative globally integrated Joule heating is compared as calculated by each model. Furthermore, hemispherically-integrated Joule heating rate estimates are compared between GITM and TIEGCM.

     This paper is organized as follows: Section 2 presents details of the GITM and TIEGCM, including their external drivers, and

describes the derivation of Joule heating in both models. Section 3 presents the results of the implementation of the simulations for St Patrick's day 2015 storm as well as the resulting Joule heating as obtained from various empirical formulations. Section 4 discusses the results, highlighting potential causes of the observed discrepancies. Finally, Section 5 summarizes the conclusions of this work.





## 2    General Circulation Models

### 2.1    The Global Ionosphere-Thermosphere Model (GITM)

GITM is a non-hydrostatic global circulation model that has been developed in order to simulate the energy balance, chemistry, and dynamics of the Earth's ionosphere and thermosphere (Ridley et al., 2006; Vichare et al., 2012; Deng et al., 2019). It has also been used to simulate planetary upper atmospheres (Bougher et al., 2015). GITM simulates the state of the mutually coupled ionosphere and thermosphere at altitudes from 100 km to ∼600 km. It solves the coupled continuity, momentum and energy equations of neutrals and ions. The continuity, momentum, and energy equations in GITM have realistic source terms. Furthermore, GITM solves for the horizontal advection for both ions and neutrals using a 2nd order Rusanov solver that make no smoothing approximations near the poles. The complete vertical momentum equation is solved using the AUSM solver (see, e.g., Ullrich et al., 2010), with each species having individual vertical momentum equations and velocities with diffusive coupling terms that limit the inter-species flows in the lower regions of the model where the Eddy diffusion is large. The vertical grid spacing is typically around 0.3 times the scale height of the dayside low latitude thermosphere and is set at the start of the simulation. The low latitude dynamo is described by Vichare et al. (2012). The chemistry is solved for implicitly, allowing for rapid variations in the individual species densities. The electron and ion temperatures are solved for using semi-implicit schemes, as described by Zhu and Ridley (2016), allowing for non-steady-state evolution of both. Each neutral species has a distinct vertical velocity, with a frictional term linking the velocities. Ion species in GITM include: $O^+(4S)$, $O^+(2D)$, $O^+(2P)$, $O_2^+$, $N^+$, $N_2^+$, and $NO^+$ whereas neutral include: $O$, $O_2$, $N(2D)$, $N(2P)$, $N(4S)$, $N_2$, and $NO$. A key advantage of GITM is that it is capable of employing a versatile, non-uniform grid, with variable resolution in both altitude and latitude, as opposed to a pressure grid that is commonly used in other thermosphere codes. The vertical grid spacing is less than 3 km in the lower thermosphere, at altitudes from 100 to ∼250 km, whereas it is over 10 km in the upper thermosphere, at altitudes from ∼250 km to 600 km. The ion momentum equation is solved with the assumption of a stable state, while accounting for the pressure, gravity, neutral breezes, and external electric fields. Several high-latitude ionospheric electrodynamic models have been used as external drivers of GITM; these include, among others, the Assimilative Mapping of Ionosphepric Electrodynamics (AMIE) approach Richmond and Kamide (1988a), the Weimer model (Weimer, 2005), and the Ridley et al. electrodynamic potential pattern (Ridley et al., 2000). GITM model runs are initiated in a number of different ways, such as (1) utilizing an ideal environment in which the user inputs the density and temperature at the base of the atmosphere; (2) using MSIS Picone et al. (2002) and International Reference Ionosphere (IRI) Bilitza (2018); and (3) starting from a prior run. In the present study, the second of the above initialization approaches is followed.

Using the geophysical parameters that are produced as outputs of GITM, Joule heating can then be estimated. These estimations require in addition the computation of electrical current $j$ and Pedersen conductivity, $\sigma_P$. The equations that are used in the estimations of the above heating rates are presented in Section 2.3; their derivations are further elaborated in Sarris et al. (2023b).





## 2.2 The Thermosphere, Ionosphere, and Electricity General Circulation Model (TIEGCM)

The NCAR Thermosphere, Ionosphere, and Electricity General Circulation Model (TIEGCM) is a first-principles, three-dimensional, nonlinear description of the linked thermosphere and ionosphere system with a self-consistent solution of the middle and low-latitude dynamo field (see, e.g., Qian et al., 2014). The three-dimensional momentum, energy and continuity

equations for neutral and ion species are solved at each time-step using a semi-implicit, fourth-order, centered finite difference method on each pressure surface in a staggered vertical grid. The main assumptions used in TIEGCM calculations include steady-state for the ion and electron energy equations, hydrostatic assumption and constant gravity. A streamlined formulation is used for eddy diffusion. Photoelectron heating is based on a streamlined connection. Simple empirical specifications define the upper boundary requirements for electron heat and flux transfer. Furthermore, TIEGCM also solves for the vertical mo-

mentum equation. Ion species in TIEGCM include: $O^+$, $O_2^+$, $N_2^+$, $NO^+$, and $N^+$ whereas neutral include: $O$, $O_2$, $N_2$, $NO$, $N(4S)$, $N(2D)$. In TIEGCM, $CO_2$ is assumed to be in diffusive equilibrium, although it is not explicitly solved. Similarly to GITM, Joule heating is subsequently estimated based on the geophysical parameters that are provided as outputs of TIEGCM. The equations that are used in the estimations of the above heating rates are further discussed in Section 2.3.

## 2.3 Derivation of Joule heating rate in TIE-GCM and GITM

In this section the methodology for calculating the Joule heating rates in GITM and TIEGCM is presented, which is slightly different between the two GCMs: Whereas GITM calculates Joule heating by calculating the complete neutral-ion collisional heating rate, as described in Killeen et al. (1984) and Zhu and Ridley (2016), TIEGCM follows the approach outlined in Lu et al. (1995). In the following, the equivalence of the two methodologies is derived, highlighting the assumptions used in each methodology. The derivation is initiated by applying the Poynting theorem to the high-latitude ionosphere:

$$\frac{\partial W}{\partial t} + \nabla \cdot \boldsymbol{S} + \boldsymbol{J} \cdot \boldsymbol{E} = 0 \tag{1}$$

where $W$ is the electromagnetic energy density, $\boldsymbol{S}$ is the Poynting vector, $\boldsymbol{J}$ is the electric current and $\boldsymbol{E}$ is the electric field. Neglecting the electromagnetic energy density rate of change by assuming a quasi-steady state, equation (1) becomes:

$$\nabla \cdot \boldsymbol{S} + \boldsymbol{J} \cdot \boldsymbol{E} = 0 \tag{2}$$

The $\boldsymbol{J} \cdot \boldsymbol{E}$ term is the energy dissipated/generated as denoted by Lu et al. (1995). By accounting that the parallel to the ambient magnetic field component of the electric field is much smaller than the perpendicular component ($\boldsymbol{E} \approx \boldsymbol{E}_\perp$), the $\boldsymbol{J} \cdot \boldsymbol{E}$ becomes $\boldsymbol{J}_\perp \cdot \boldsymbol{E}_\perp$.

Ionospheric Joule heating is calculated in the reference frame of the neutral constituents. Thus, by assuming that the neutrals move with a velocity $\boldsymbol{u}_n$, the electric field in the reference frame of the neutrals is expressed as:

$$\boldsymbol{E}_\perp^* = \boldsymbol{E}_\perp + \boldsymbol{u}_n \times \boldsymbol{B} \tag{3}$$





Thus,

$$\boldsymbol{E}_\perp = \boldsymbol{E}_\perp^* - \boldsymbol{u}_n \times \boldsymbol{B} \tag{4}$$

By using equation (4), the electromagnetic energy exchange rate becomes:

$$\boldsymbol{J}_\perp \cdot \boldsymbol{E}_\perp = \boldsymbol{J}_\perp \cdot \boldsymbol{E}_\perp^* - \boldsymbol{J}_\perp \cdot (\boldsymbol{u}_n \times \boldsymbol{B}) \tag{5}$$

where the term $\boldsymbol{J}_\perp \cdot \boldsymbol{E}_\perp^*$ is the Joule heating rate and the term $\boldsymbol{J}_\perp \cdot (\boldsymbol{u}_n \times \boldsymbol{B})$ is the mechanical energy transfer to the neutrals
Lu et al. (1995). Thus, the Joule heating rate can be expressed as:

$$q_{JH} = \boldsymbol{J}_\perp \cdot \boldsymbol{E}_\perp^* \tag{6}$$

Regarding the electrical current term, applying Ohm's to the ionospheric plasma leads to:

$$\boldsymbol{J}_\perp = \boldsymbol{J}_P + \boldsymbol{J}_H = \sigma_P \boldsymbol{E}_\perp^* - \sigma_H (\boldsymbol{E}_\perp^* \times \hat{b}) \tag{7}$$

where $\boldsymbol{J}_P$ is the Pedersen current, $\boldsymbol{J}_H$ is the Hall current, $\hat{b}$ is the unit vector among the ambient magnetic field, and $\sigma_P$ and $\sigma_H$ are the Pedersen and Hall conductivities respectively. The Hall current is non-dissipative, and the power transfer is achieved by the Pedersen current; thus, equation (5) becomes:

$$q_{JH} = \boldsymbol{J}_P \cdot \boldsymbol{E}_\perp^* = (\sigma_P \boldsymbol{E}_\perp^*) \cdot \boldsymbol{E}_\perp^* = \sigma_P |\boldsymbol{E}_\perp + \boldsymbol{u}_n \times \boldsymbol{B}|^2 \tag{8}$$

Equation (8) is the expression used internally by TIEGCM for the calculation of Joule heating in the model.

As discussed above, GITM follows a different approach in calculating Joule heating, by calculating the complete neutral-ion collisional heating rate, given as in Killeen et al. (1984) and Zhu and Ridley (2016):

$$q_{JH} = \sum_n n_n m_n \sum_i \frac{\nu_{ni}}{m_i + m_n} [3k_B(T_i - T_n) + m_i(\boldsymbol{u}_n - \boldsymbol{v}_i)^2] \tag{9}$$

where $n_n$ is the neutral number density, $m_n$ is the neutral mass, $m_i$ is the ion mass, $\nu_{ni}$ is the neutral-ion collision frequency, $k_B$ is the Boltzmann constant, $T_i$ and $T_n$ are the ion and neutral temperatures respectively and $v_i$ is the ion velocity.
Subsequently, the equivalence of (8) and (9) with respect to the calculation of Joule heating rates in the ionosphere needs to be shown. By assuming that the ion temperature is in steady state and that the ions are coupled to both the neutrals and electrons, the ion energy equation is derived as:





$$3k_B N_e \frac{m_i}{m_i + m_n} \nu_{in}(T_i - T_n) = N_e \nu_{in} \frac{m_i m_n}{m_i + m_n}(\boldsymbol{u}_n - \boldsymbol{v}_i)^2 +$$

$$3k_B N_e \frac{m_i}{m_i + m_e} \nu_{ie}(T_e - T_i) + N_e \nu_{ie} \frac{m_i m_e}{m_i + m_e}(\boldsymbol{u}_e - \boldsymbol{v}_i)^2 \quad (10)$$

Considering $m_e << m_i$, thus $m_i/(m_i + m_e) \approx 1$ and after some manipulations, equation (10) becomes:

$$3k_B \frac{m_i}{m_i + m_n}(T_i - T_n) = \frac{m_i m_n}{m_i + m_n}(\boldsymbol{u}_n - \boldsymbol{v}_i)^2 +$$

$$3k_B \frac{\nu_{ie}}{\nu_{in}}(T_e - T_i) + \frac{\nu_{ie}}{\nu_{in}} m_e (\boldsymbol{u}_e - \boldsymbol{v}_i)^2 \quad (11)$$

Collisions between electrons and ions become important (compared to ion-neutral collisions) only in the upper ionosphere,
where, however, ions and electrons have almost similar velocities perpendicular to the ambient magnetic field ($E \times B$ drift),
thus $\boldsymbol{v}_{i\perp} - \boldsymbol{v}_{e\perp} \approx 0$. Furthermore, in general, at high latitudes, $\nu_{ie} << \nu_{in}$, and thus (11) becomes:

$$3k_B(T_i - T_n) \approx m_n(\boldsymbol{u}_n - \boldsymbol{v}_i)^2 \quad (12)$$

By substituting (12) into (9) we get:

$$q_{JH} = \sum_n n_n m_n \sum_i \frac{\nu_{ni}}{m_i + m_n}[m_n(\boldsymbol{u}_n - \boldsymbol{v}_i)^2 + m_i(\boldsymbol{u}_n - \boldsymbol{v}_i)^2] \quad (13)$$

Finally, using the relation between ion-neutral and neutral-ion collision frequencies:

$$n_n m_n \nu_{ni} = n_i m_i \nu_{in} \quad (14)$$

equation (13) becomes:

$$q_{JH} = \sum_i n_i m_i \sum_n \nu_{in}(\boldsymbol{u}_n - \boldsymbol{v}_i)^2 \quad (15)$$

which is the ion-neutral frictional heating rate. The equivalence between the ion-neutral frictional heating rate and the Joule
heating rate has been proven in Strangeway (2012), and thus the equivalence of the Joule heating calculation between GITM
and TIEGCM is derived.

The Pedersen conductivity that is needed for the calculation of Joule heating in equation (8) is calculated as:

$$\sigma_P = \frac{q_e}{B}\left[N_{O^+}\frac{r_{O^+}}{1 + r_{O^+}^2} + N_{O_2^+}\frac{r_{O_2^+}}{1 + r_{O_2^+}^2} + N_{NO^+}\frac{r_{NO^+}}{1 + r_{NO^+}^2} + N_e\frac{r_e}{1 + r_e^2}\right] \quad (16)$$





where $r_{O^+}$, $r_{O_2^+}$, $r_{NO^+}$ and $r_e$ are the collision to gyrofrequency ratios (i.e. $\nu_{i(e)n}/\Omega_{i(e)}$) of $O^+$, $O_2^+$, $NO^+$ and $e$ respectively, which are calculated as described in tables 4.4 and 4.5 of Schunk and Nagy (2009), and $N_{O^+}$, $N_{O_2^+}$, $N_{NO^+}$ and $N_e$ are the number densities of species in $m^{-3}$. Collision frequencies of the aforementioned species are calculated for collisions with neutral species of $O$, $O_2$ and $N_2$.

In order to calculate the global heating rates over the same altitude range in the two GCMs, the outputs of each of the two GCMs are integrated in altitude from 100km to 600km, and across all geographic latitudes and longitudes. Such altitude-integrated Joule heating rates have also been calculated in a number of prior studies, such as by Lu et al. (1995); Thayer (1998); Weimer (2005) and Deng et al. (2009). In this study, height integration is performed based a trapezoidal integration scheme, according to:

$$\int_a^b f(z)dz = \sum_{k=1}^{N} \frac{f(z_k - 1) + f(z_k)}{2} \Delta z \tag{17}$$

where $f$ denotes the altitude-resolved quantity that is integrated, $z$ is the altitude, are the $a$ and $b$ are the upper and lower limits of integration respectively and $k$ denotes the provided discrete altitude levels.

Further details on the analysis presented herein can be found in, e.g., Sarris et al. (2020), Sarris et al. (2023b) and references therein. The above calculations were performed using the integration module of the open-source code DaedalusMASE (Sarris et al., 2023b), which has been translated to C++ from the original code that was written in python so as to be more efficient in terms of execution time.

## 3 Simulations

### 3.1 St Patrick's day storm

GITM and TIEGCM runs, as well as calculations based on empirical formulations, were performed for St Patrick's day storm of March 2015, which is the first and also the largest geomagnetic storm of solar cycle 24. Various aspects of this storm have been described in numerous studies, including, for example, the work of Kanekal et al. (2016) and Hudson et al. (2017) who studied the prompt injection and acceleration of energetic electrons, Jaynes et al. (2018) and Ozeke et al. (2019) who investigated the fast radial diffusion driven by ULF waves, Lyons et al. (2016), Marsal et al. (2017) and Prikryl et al. (2016) who studied ionospheric disturbances induced by energy inputs into the high-latitude regions, Wei et al. (2019), Zhang et al. (2017) and Yue et al. (2016) who studied sub-auroral processes related to magnetosphere-ionosphere coupling, Dmitriev et al. (2017) and Zakharenkova et al. (2016) who studied changes in global neutral wind driven by high-latitude energy and momentum inputs, and Zhang et al. (2022a) who focused on the generation and propagation of the induced electric field that was responsible for the prompt acceleration of energetic electrons during this storm. In this study, we estimate the total Joule heating dissipation during this event, and we investigate discrepancies between GITM and TIEGCM when driven with different electric field specifications and auroral precipitation models; we also compare these results against various commonly used empirical models.





An overview of St Patrick's day storm of March 2015 is presented in the top panels of Figure 1. The storm was caused by a coronal mass ejection that arrived at Earth on March 17 at ∼04:45 UT, whereas the main phase of the storm began at ∼06:00

UT, indicated by the first vertical dashed line marked as A, when the $Dst$ index started to gradually decrease (Figure 1 panel (a)) and the $B_z$ component of the interplanetary magnetic field (IMF) turned southward for the first time during this event (Figure 1 panel (b)). Between ∼06:00 UT and ∼12:20 UT, indicated by the second vertical dashed line marked as B, the IMF $B_z$ alternated between northward and southward, whereas after ∼12:20 UT it turned southward and remained that way until the next day. The $Dst$ index continued to decrease, reaching its minimum of -223 nT on 17 March at ∼23:20 UT. This was

followed by a long recovery phase. The planetary $Kp$ index, also shown in Figure 1 panel (a), reached its maximum value of 7+ to 8- from ∼12 UT to 24 UT.

### 3.2 Model Drivers and Inputs

As described above, GITM and TIEGCM are externally driven by the specification of electric fields and auroral precipitation. In this study the following four different runs are performed and inter-compared:

1. GITM with the Weimer electric field model and the Feature Tracking of Aurora (FTA) Wu et al. (2021) empirical model, hereafter referred to as $GITM_{Weimer}$

    2. TIEGCM with the Weimer electric field model and the Emery et al. (2012) empirical auroral model, hereafter referred to as $TIEGCM_{Weimer}$

    3. GITM with the AMIE data assimilation model for both the electric fields and auroral inputs, hereafter referred to as
220        $GITM_{AMIE}$

    4. TIEGCM with the AMIE data assimilation model for both the electric fields and auroral inputs, hereafter referred to as $TIEGCM_{AMIE}$

Runs 1 and 2 utilize the same electric field specification based on the Weimer model, albeit employing different auroral models. It is noted that these configurations represent the default (and thus more commonly used) setups for the two models,

and hence identifying differences in the estimates of Joule heating during active times is of importance. In further detail, with respect to Run 1, GITM uses a two-step process to model auroral inputs: in a first step, the empirical model of Wu et al. (2021) is used to specify the auroral oval latitude, local time maps of the average energy and energy flux; subsequently, taking the average energy, energy flux and mass density of the thermosphere, the ionization rate is calculated as a height profile throughout the thermosphere/ionosphere using the approach described in Sharber et al. (1996). Runs 3 and 4 utilize identical

high-latitude drivers, both for the electric field specification and auroral input, and can thus be directly inter-compared. In terms of their initialization, GITM was run for a duration of 24 hours prior to the onset of the solar storm. This period allowed for the stabilization of the model in terms of density, wind, and temperature outputs. TIEGCM was initiated with a history file dated March 15, 2015, marking the start of this simulation.





It is noted in particular that the Weimer 2005 model is the default electric field specification for both GCMs and relies
on certain Interplanetary Magnetic Field (IMF) parameters as input, including plasma density, solar wind velocity $V_x$ (in the
Sun-Earth direction), and the perpendicular orientation of the solar wind magnetic field $B_y$ and $B_z$. The AMIE procedure is a
data assimilation method which provides maps of high-latitude electric fields, currents, and the associated magnetic variations
based on collections of localized observational data. In addition to IMF parameters, both GITM and TIEGCM use as input the
daily $F10.7$ index, an 81-day average of $F10.7$ and the 3-hourly $Kp$ index. It is noted that TIEGCM uses the above inputs with
a 15-min resolution, but calls the IMF data every 1 minute, whereas GITM uses all the above inputs with a 1 minute resolution.
Moreover, GITM requires as input the maximum eastward auroral electrojet strength ($SMU$), the maximum westward auroral
electrojet strength ($SML$) and the difference between the two ($SME$). $SMU$, $SML$ and $SME$ are referred to as SuperMAG
indices and are analogous to $AU$, $AL$ and $AE$; they have been introduced as high spatial resolution alternatives to $AU$, $AL$
and $AE$ Gjerloev (2012); Bergin et al. (2020) and are used herein to drive FTA model.

Panels (a) through (g) of Figure 1 present the aggregated driving inputs of GITM and TIEGCM, as described above, as well
as the indices used as inputs for the empirical parameterizations of Joule heating, as follows: Panel (a) presents the $Dst$ index
(green color), the $SYM-H$ index (dark-cyan color), the 3-hourly $Kp$ index (purple) and the daily $F10.7$ index (blue dashed
line). Panel (b) shows the $AL$ index (orange) and the $AE$ index (cyan). Panel (c) shows the IMF components, $B_y$ (blue) and
$B_z$ (brown), in Geocentric Solar Magnetospheric (GSM) coordinates, for the duration of St Patrick's day storm; the vertical
line marked as A indicates the first southward turning of $B_z$, indicating the start of the main phase of the storm, while vertical
line B in the same figure indicates the start of a prolonged period when $B_z$ remains southward; this is further discussed below.
Panel (d) presents the solar wind velocity $V_x$ along the Sun-Earth line (blue solid line) and the plasma density, in units of
n/cc (brown solid line). Panel (e) presents the maximum eastward auroral electrojet strength (blue solid line), the maximum
westward auroral electrojet strength (blue dashed line) and the difference between the two (brown solid line), which are used
in driving the GITM model in addition to the inputs shown in panels (a), (d) and (e).

### 3.3 Empirical Formulations

Further to the calculation of Joule heating rates in GITM and TIEGCM, ionospheric dissipation through Joule heating are com-
monly approximated via empirical formulations that use geomagnetic indices as input. Several studies have derived empirical
relationships for the quantification of hemispheric and global Joule heating that are using the AE or AL indices as inputs; these
include the studies by Perreault and Akasofu (1978), Akasofu (1981), Ahn et al. (1983), Baumjohann and Kamide (1984),
Cooper et al. (1995), Lu et al. (1995). Later on, Chun et al. (1999) estimated Joule heating with a quadratic fit to the Polar
Cap ($PC$) index. Expanding upon the work of Chun et al. (1999), Knipp et al. (2005) proposed an empirical formula based on
the $PC$ and the Disturbance Storm Time ($Dst$) indices. Moreover, Weimer (2005) proposed another method to estimate Joule
heating; it is noted that in the study of Weimer (2005) Joule heating and Poynting flux are used interchangeably. A summary
of the above studies and the corresponding relationships as well as constraints in terms of season or hemisphere where these
are applicable are presented in Table 1.





**Table 1.** Empirical Formulas for Joule Heating Estimations

| Study | Formula | Hemisphere | Season |
|---|---|---|---|
| Perreault and Akasofu (1978) | $0.05AE(12)$ | - | - |
| Akasofu (1981) | $0.1AE(12)$ | N | Spring |
| Ahn et al. (1983) | $0.23AE(12)$ | N | Spring |
| Ahn et al. (1983) | $0.19AE(71)$ | N | Spring |
| Ahn et al. (1983) | $0.3AL(12)$ | N | Spring |
| Ahn et al. (1983) | $0.27AL(71)$ | N | Spring |
| Baumjohann and Kamide (1984) | $0.32AE(12)\pm 5$ | N | Spring |
| Baumjohann and Kamide (1984) | $0.33AE(71)\pm 5$ | N | Spring |
| Baumjohann and Kamide (1984) | $0.4AL(71)\pm 5$ | N | Spring |
| Cooper et al. (1995) | $0.54AE(12)-49$ | N | Autumn |
| Cooper et al. (1995) | $0.28AE(AMIE)-20$ | N | Autumn |
| Lu et al. (1995) | $0.33AE(12)-26$ | N | Spring |
| Chun et al. (1999) | $4.14PC^2+25PC+8.9$ | - | Equinox |
| Knipp et al. (2005) | $2.54PC^2+29.14PC+0.21Dst+0.0023Dst^2$ | - | - |
| Weimer (2005) | $E\times\Delta B/\mu_0$ | - | - |

*The numbers in parentheses indicate the number of magnetic stations used in the study

The $AE$ and $AL$ indices were used in the first twelve empirical formulations of Table 1 with a 1-minute resolution, and were obtained from the World Data Center (WDC) for Geomagnetism, Kyoto, Japan. The Polar Cap index, also used with a 1-minute resolution, consists of the Polar Cap North ($PCN$) index and the Polar Cap South ($PCS$) index. $PCN$ index is taken from the National Space Institute, Technical University of Denmark (DTU, Denmark) and PCS index from the Arctic and Antarctic Research Institute (AARI, Russian Federation). The $Dst$ index was obtained with a 1-hour resolution from WDC, Kyoto, Japan. In order to calculate Joule heating according to Knipp et al. (2005) with a 1-minute resolution, we replaced the 1-hour $Dst$ index with the $SYM-H$ index at 1 min resolution from WDC, Kyoto, Japan; as discussed in Wanliss and Showalter (2006), the $Dst$ and $SYM-H$ indices are considered equivalent but with different time resolutions. A comparison between the two indices is presented in Figure 1 panel (a). The datasets used in this study are readily available at **?**.

## 3.4 Model Runs

In terms of resolution of the two GCMs, the TIEGCM run was performed with a spatial resolution of 2.5 degrees in latitude and longitude, 4 grid points per scale height and a time step of 30 seconds. The GITM run was performed with a resolution of 2 degrees in latitude and 4 degrees in longitude. The altitude resolution of GITM is 3 grid points per scale height and the temporal resolution is 10 seconds. The resulting output datasets were then converted to a common format for further processing. The





datasets and the code are available though **?**. Models Runs were performed on a CPU-based machine with 64GB RAM and an Intel(R) Core(TM) i9-9900K CPU @ 3.60GHz.

In Figure 1 panel (g) the globally-integrated Joule heating rates are presented as calculated using the four GCM runs and the empirical models, as follows: The Joule heating rates as calculated according to the (i) $GITM_{Weimer}$, (ii) $GITM_{AMIE}$,

(iii) $TIEGCM_{Weimer}$ and (iv)$TIEGCM_{AMIE}$ runs are marked, respectively, with (i) a thicker dark blue line; (ii) a thicker dark purple line; (iii) a thicker brown line; and (iv) a thicker green line; and Joule heating rates as estimated according to the various empirical formulations of Table 1 are plotted with thinner lines, as marked in the inset of the figure, in chronological order. It is noted that several of the empirical formulations listed in this table give hemispheric estimates of Joule heating. In order to compare against the results presented in Figure 1, these were multiplied by a factor of 2 to obtain approximations of

the global values of Joule heating. Additionally, Figure 1 panel (f) presents the hemispheric power (HP) as it is calculated by the four GCM runs, as marked.

In order to investigate the inter-hemispheric asymmetries of Joule heating, in Figure 2 the integrated Joule heating is plotted separately over the Northern and Southern hemispheres, in panels (a) and (b), respectively. The ratio between Joule heating in the northern hemisphere over Joule heating in the southern hemisphere ($NH/SH$) is plotted separately for each of the

four GCM runs as follows: in panel (c), $NH/SH$ is plotted with solid lines for $GITM_{Weimer}$ (blue) and $TIEGCM_{Weimer}$ (brown); and in panel (d), $NH/SH$ is plotted with dashed lines for $GITM_{AMIE}$ (blue) and $TIEGCM_{AMIE}$ (brown).

In order to cross-compare the total amount of Joule heating that is deposited onto each thermospheric hemisphere during St Patrick's day storm 2015 as estimated by the two GCMs and the various empirical models, in Figure 3 the cumulative, time-integrated Joule heating is plotted as a function of time. The corresponding models are color-coded and are listed in the

right-hand side of the figure in order of descending Joule heating. The estimated cumulative Joule heating in the northern (southern) hemisphere are plotted in GITM and TIEGCM with thicker solid (dashed) lines. The thinner lines indicate Joule heating estimates over the northern hemisphere according to the empirical models of Table 1, as marked in the figure's inset. In the cases that empirical estimates are based on indices obtained from 12 ground stations, the results are plotted with a thin solid line, whereas estimates that are based on 71 stations are plotted with a thin dotted line.

In Figure 4 three snapshots of the height-integrated Joule heating is shown as a polar plots over the northern hemisphere, based on $GITM_{Weimer}$ (panel (a)) and $TIEGCM_{Weimer}$ (panel (b)) simulations. Three characteristic times during St Patrick's day storm on 17 March 2015 are plotted: 06:20 UT (left-hand side panels), corresponding to the beginning of the storm; 14:10 UT (middle panels), corresponding to the time of maximum percentage difference in Joule heating between $GITM_{Weimer}$ and $TIEGCM_{Weimer}$; and 22:50 UT (right-hand side panels), corresponding to line D in the above figures,

which marks the peak of the storm, as indicated by the minimum in $Dst$. In Figure 4 the height-integrated Joule heating is plotted for the same time-steps as those presented in panels (a) and (b), but based on $GITM_{AMIE}$ (panel (c)) and $TIEGCM_{AMIE}$ (panel (d)). The comparisons between Figure 4 panels (a) and (c) the top and panels (b) and (d) show that, apart for the large differences in the amplitudes of Joule heating between GITM and TIEGCM, the distribution of Joule heating in longitude and latitude shows notable similarities, with slight variations in the localization and extent of the spatial structures where Joule

heating appears.



**Figure 1.** Joule Heating in combination with Geophysical Indices and used quantities. (a) $Dst$ (green), $SYM-H$ (cyan), $F10.7$ (dashed blue) and $Kp$ (purple) space indices. (b) $AL$ (orange) and $AE$ (cyan). (c) $B_y$ (dashed blue) and $B_z$ (brown) IMF components. (d) $V_x$ (blue) and $Plasma\ Density$ (brown). (e) $SME$ (cyan), $SMU$ (blue) and $SML$ (orange). (d) Hemispheric Power from GCMs, HP from $GITM_{Weimer}$ northern hemisphere (blue), HP from $GITM_{Weimer}$ southern hemisphere (dashed orange), HP from $GITM_{AMIE}$ northern hemisphere (green), HP from $GITM_{AMIE}$ southern hemisphere (red), HP from $TIEGCM_{Weimer}$ same for each hemisphere (purple) and HP from $TIEGCM_{AMIE}$ same for each hemisphere (brown). (g) Joule Heating from various GCMs and Emprical models as marked.



**Figure 2.** Time-series of the hemisphericaly-integrated Joule Heating in (a) the Northern Hemisphere (NH) and (b) the Southern Hemisphere (SH). (c) Percentage difference between NH & SH of $GITM_{Weimer}$ (blue solid line) and $TIEGCM_{Weimer}$ (brown solid line), and (d) percentage difference between NH & SH of $GITM_{AMIE}$ (blue dashed line) and $TIEGCM_{AMIE}$ (brown dashed line).



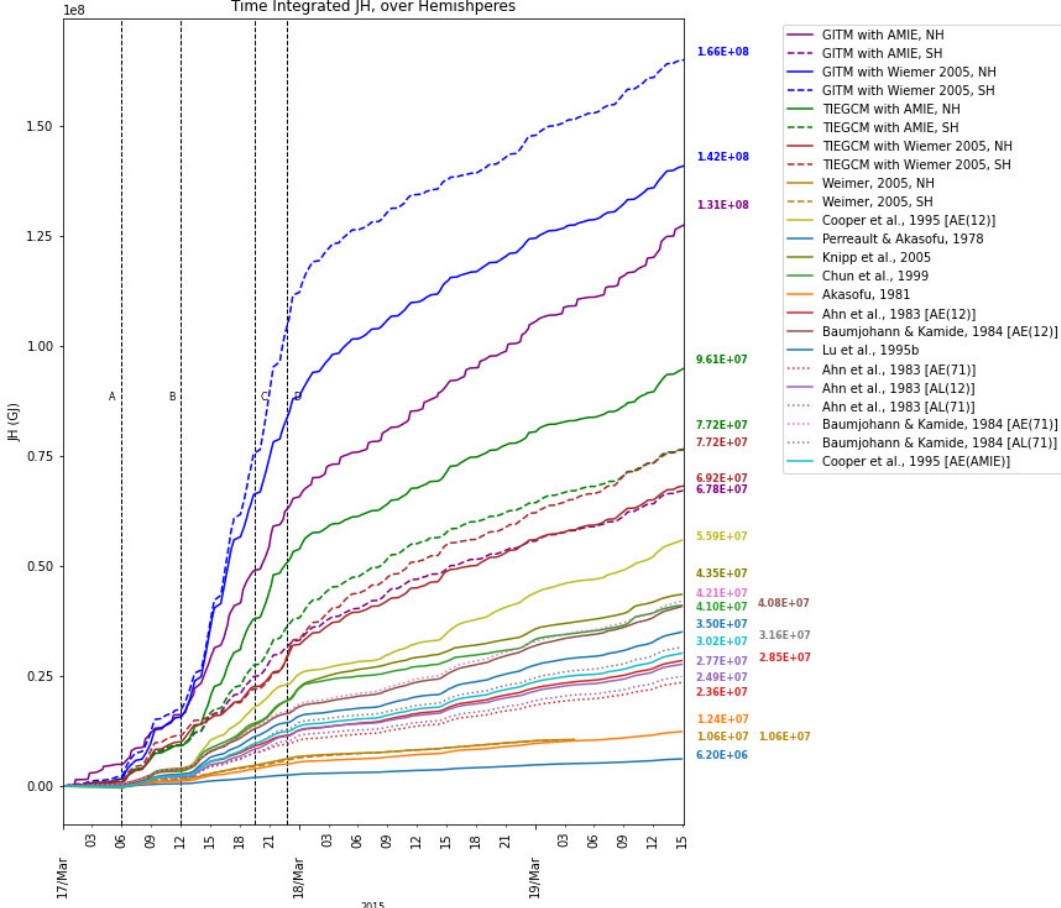

**Figure 3.** Time-integrated (cumulative) global Joule heating according to GITM and TIEGCM driven by the Weimer and AMIE high latitude electric field specifications and various empirical models, as marked, listed from highest to lowest Joule heating values.

## 4 Discussion

Based on the simulation results shown in Figures 1 and 2, a significant disagreement is observed in the values of the globally-integrated Joule heating rates as obtained through TIEGCM and GITM during the storm main phase, and in particular between March 17, 06 UT, which marks the first southward turning of $B_z$ and is noted with line A in the above figures, and March 17, 22:50 UT, which is the time of minimum $Dst$, and is marked with line D. This disagreement is noted both in terms of amplitude as well as in terms of the overall shape and evolution of Joule heating, and is more prominent when the Weimer 2005 model is used for the specification of the high-latitude electric field model, whereas a better agreement between the two GCMs is found when the AMIE model is used. A general order-of-magnitude agreement is observed in the first part of the storm, from March 17, 00 UT until 12 UT, as well as after the time of minimum $Dst$ and in the recovery phase of the storm.





**Figure 4.** Height-integrated Joule Heating as calculated over the northern hemisphere for three different snapshots during St. Patricks day event, as marked. (a) $GITM_{Weimer}$, (b) $TIEGCM_{Weimer}$, (c) $GITM_{AMIE}$, (d) $TIEGCM_{AMIE}$.




**Table 2.** Correlation coefficients between the $GITM_{Weimer}, TIEGCM_{Weimer}, GITM_{AMIE}$ and $TIEGCM_{AMIE}$ runs and their main input parameters

|  | $SME$ | $AE$ | $B_y$ | $B_z$ | $Plasma\ Den$ | $Dst$ | $SYM-H$ | $V_x$ |
|---|---|---|---|---|---|---|---|---|
| $GITM_{Weimer}$ | 0.80 | 0.34 | 0.42 | -0.24 | 0.49 | -0.56 | 0.34 | 0.44 |
| $TIEGCM_{Weimer}$ | 0.53 | 0.25 | 0.31 | -0.23 | 0.38 | -0.54 | 0.28 | 0.39 |
| $GITM_{AMIE}$ | 0.86 | 0.29 | 0.49 | -0.29 | 0.49 | -0.51 | 0.29 | 0.42 |
| $TIEGCM_{AMIE}$ | 0.85 | 0.30 | 0.48 | -0.29 | 0.49 | -0.57 | 0.30 | 0.42 |

Further to the models that have been used in this study, Suji and Prince (2018) used the OpenGGCM Raeder et al. (2001) to calculate the global ionospheric Joule heating during St. Patrick's day 2015 geomagnetic storm. The values of global Joule heating rates that are presented in Figure 7 of Raeder et al. (2001) are considerably higher than the values reported herein. Such disagreements in the comparisons of simulated Joule heating have been discussed extensively by, e.g. Cosgrove and Codrescu (2009), who investigated the underestimation of Joule heating caused by high-latitude electric field variability in electric field

models and addresses the notion of "electric field variability" as a potential source of this underestimation. Similarly, Lu et al. (2023) used TIEGCM driven with AMIE to gain insights into the 2015 St. Patrick's Day storm's impact on the ionosphere-thermosphere (IT) system by utilizing observations of high-latitude forcings, specifically aurora and electric fields, along with the TIEGCM. Verkhoglyadova et al. (2017), utilized GITM along with empirical models and proxies derived from in situ measurements to estimate the energy distribution within the IT system during the solar storms events of 16-19 March of 2013

and 2015, while Zhu and Ridley (2016), calculated Joule heating rates through GITM simulations and revealed that the globally averaged thermospheric temperature ($T_n$) was underestimated under quiet geomagnetic conditions.

In order to identify the key driving parameters for the discrepancies in Joule heating between the different model runs, a Spearman's Rank Correlation analysis Spearman (1904) has been performed between the four Joule heating rate time series and each of the input parameter time series that are shown in panels (a) through (e) of Figure 1. The results are shown

comprehensively in Table 2. Through this correlation analysis, it is found that Joule heating is strongly driven by the $SME$ electrojet strength in both the $GITM_{AMIE}, TIEGCM_{AMIE}$ and $GITM_{Weimer}$, with correlation coefficients of 0.86, 0.85 and 0.80 respectively; in $TIEGCM_{Weimer}$, this correlation drops to 0.53. The correlation coefficients of the four runs with $B_y$ are similar, but slightly smaller in $TIEGCM_{Weimer}$, whereas the correlation coefficients of the four runs with $B_z$ are similar but negative, indicating anti-correlations, with, again, a smaller correlation (in absolute value) in $TIEGCM_{Weimer}$. The anti-

correlation between Joule heating and $B_z$ is attributed to the enhanced Joule heating during southward turnings of the IMF; the dependence of Joule heating on $B_z$ has been examined in more detail in various studies, such as, e.g., by McHarg et al. (2005). The correlation with solar wind plasma density is lower for $TIEGCM_{Weimer}$ and similar between the other three runs. The correlation with the absolute value of $Dst$ is similar among all four model runs, and is the second most significant correlation, after $SME$. Finally, the correlation with $SYM-H$ is similar for all four model runs, and is comparable to the correlation with

$AE$ and $B_z$.



In Figures 1 and 2 a notable difference can be seen in Joule heating when it is calculated in $TIEGCM_{Weimer}$ compared to $GITM_{Weimer}$. This is particularly evident in the period from $\sim$12:00UT to $\sim$19:30UT on March 17, 2015, which is shown in the gray-shaded region that is bounded by lines B and C. During this time, it can be seen that $GITM_{Weimer}$ is well correlated with the $SME$ electrojet strength, with an increase and subsequent decrease in $SME$ being accompanied by a cor-
responding increase followed by a gradual decrease in Joule heating, whereas, in contrast, Joule heating in $TIEGCM_{Weimer}$ shows an initial drop followed by a gradual increase. This increase appears to be correlated with the prolonged southward turning of IMF $B_z$ during this time, which does not appear to affect in the same way the calculations of Joule heating in $GITM_{Weimer}$. Furthermore, Joule heating as computed by $GITM_{Weimer}$ exhibits a higher magnitude compared to the results from $TIEGCM_{Weimer}$, as well as the results from $GITM_{AMIE}$ and $TIEGCM_{AMIE}$. This is attributed to the differences
in the auroral precipitation model that is used for the two runs that utilize the Weimer 2005 model as input, $GITM_{Weimer}$ and $TIEGCM_{Weimer}$, as is also indicated by the differences in the HP power presented in Figure 1 panel (f). As discussed above, the $GITM_{Weimer}$ run uses the FTA model Wu et al. (2021), while $TIEGCM_{Weimer}$ run uses the analytical auroral model of Roble and Ridley (1987) and Emery et al. (2012). On the other hand, $GITM_{AMIE}$ and $TIEGCM_{AMIE}$ use the same auroral precipitation model, described in Richmond and Kamide (1988a), and show better agreement in terms of shape and magnitude,
as shown in Figure 1 panel (g), especially between lines B and C. It is also noted that in the $TIEGCM_{Weimer}$ run a much lower variability is observed in Joule heating compared to the other three runs, which have a larger peak-to-peak fluctuation in the amplitudes of Joule heating. It is speculated that this is due to the lower level of correlation between the $TIEGCM_{Weimer}$ and the $SME$ index, compared to the other three runs, as discussed above and as shown in Table 2. Finally, it is noted that all empirical formulations tend to underestimate the overall Joule heating when compared to all four GCM runs, as shown in
Figure 1 panel (g).

Comparing the time series of the hemispherically-integrated Joule heating from the two GCMs with the corresponding values from the empirical models, as is plotted in Figure 2 panel (a), it can be seen that there is a closer agreement between the empirical models and $TIEGCM_{Weimer}$ rather than with $GITM_{Weimer}$. Furthermore, comparing the percentage differences between the hemispherically-integrated Joule heating in the northern and southern hemispheres as calculated with
$GITM_{Weimer}$ and $TIEGCM_{Weimer}$, which are plotted in Figure 2 panel (c), it can be seen that the two model runs show significantly different inter-hemispheric asymmetries, especially during times of enhanced Joule heating, between lines B and D. During this time, $GITM_{Weimer}$ calculations show higher Joule heating in the southern hemisphere by up to $\sim 55\%$, while $TIEGCM_{Weimer}$ shows initially higher Joule heating in the northern hemisphere by up to $\sim 35\%$ and subsequently higher Joule heating in the southern hemisphere by up to $\sim 40\%$, with a time-lag of approximately 5 hours. The percentage differ-
ences between the hemispherically-integrated Joule heating from $GITM_{AMIE}$ and $TIEGCM_{AMIE}$ are presented in Figure 2 panel (d) and show almost the same inter-hemispheric asymmetry during the simulation period. Both $GITM_{AMIE}$ and $TIEGCM_{AMIE}$ show higher Joule heating in the northern hemisphere by up to $\sim 116\%$ and $\sim 100\%$ respectively during the storm main phase, while during the recovery face of the storm Joule heating deposition in the southern hemisphere becomes higher by up to $\sim 60\%$ for the $GITM_{AMIE}$ and up to $\sim 74\%$ for the $TIEGCM_{AMIE}$.



By comparing the time-integrated (cumulative) hemispherically-integrated Joule heating, as shown in Figure 3, it can be seen that, whereas higher Joule heating is observed in the southern hemisphere (SH; dashed lines) than the northern hemisphere (NH; solid lines) for the $GITM_{Weimer}$ (blue) and $TIEGCM_{Weimer}$ (brown) simulations, the opposite is observed in the case of $GITM_{AMIE}$ (purple) and $TIEGCM_{AMIE}$ (green), with the northern hemisphere receiving larger amounts of heating over the course of the storm.

Such asymmetries in the hemispherically-integrated Joule heating have been identified by several studies, and have been associated with the Earth's asymmetric magnetic field configuration: as discussed in, e.g., Laundal et al. (2017) and references therein, the dipole tilt and eccentricity shift in the Earth's magnetic field leads to a displacement between the geographic and geomagnetic poles, which is larger in the southern hemisphere, and to a difference in the magnetic field strength between north-south conjugated latitudes. Hong et al. (2021) used GITM to study the impacts of different causes on the inter-hemispheric

asymmetry of the ionosphere-thermosphere system, including inter-hemispheric differences associated with the solar irradiance, the geomagnetic field, and the magnetospheric forcing under moderate geomagnetic conditions. Hong et al. (2021) derived an index of inter-hemispheric asymmetry for Joule heating, which, for solar equinox conditions, was found to be as large as $\sim 43\%$ due to the asymmetric geomagnetic field, $\sim 28\%$ due to asymmetric particle precipitation and $\sim 35\%$ due to asymmetric ion convection pattern. Pakhotin et al. (2021), using Swarm satellite observations, demonstrated that the northern

hemisphere generally receives a higher electromagnetic energy input across all seasons. This preference has also been observed using DMSP satellites Knipp et al. (2021). Cosgrove et al. (2022) also investigated the appearance of such asymmetries, and revealed a considerable asymmetry in the hemispherically integrated Poynting flux between the northern hemisphere (NH), which generally showed higher flux, and the southern hemisphere (SH). More recently, Smith et al. (2023), also investigated inter-hemispheric asymmetries, looking into the role of solar wind driving conditions and the accuracy of Joule heating es-

timates using GITM (Weimer and AMIE) runs during the 2013 St. Patrick's Day geomagnetic storm (note the different year compared to the 2015 St. Patrick's Day geomagnetic storm that was simulated herein). They showed that AMIE driven simulations lead to stronger inter-hemispheric asymmetries in Joule heating compared to Weimer 2005 driven runs, and found higher Joule heating deposition in the southern hemisphere for the first phase of the 2013 St. Patrick's Day geomagnetic storm for both Weimer and AMIE simulation, which reversed in later phases of that storm.

The results presented herein indicate that the appearance of inter-hemispheric asymmetries are largely dependent on the external drivers that are used (primarily the electric field and auroral precipitation specifications). The results show that both the $GITM_{AMIE}$ and $TIEGCM_{AMIE}$ runs have almost the same inter-hemispheric asymmetry and Joule heating magnitude, as observed in Figure 2 panel (d), denoting that the implementation of the two GCMs delivers almost the same results under the same driving conditions (electric field and auroral precipitation). This is not confirmed in the case of the $GITM_{WEIMER}$ and

$TIEGCM_{WEIMER}$ runs in their default configuration, as observed in Figure 2 panel (c), where the Joule heating deposition is largely variable, as discussed above. Taking into account that both of these runs use the Weimer 2005 model as a high latitude electric field driver, these results show that Joule heating is highly dependent on the auroral precipitation model.

     It is noted that, as discussed above, both GITM and TIEGCM use the International Geomagnetic Reference Field (IGRF) magnetic field model, and hence the asymmetries in the magnetic field are the same; thus the differences in the observed





behavior are more likely attributed to the asymmetric particle precipitation and the asymmetric ion convection pattern. However the exact causes of the different behavior of TIEGCM and GITM when run using different external drivers with respect to the inter-hemispheric differences is a subject that requires further research through parametric studies.

## 5   Summary and Conclusions

Based on GITM and TIEGCM driven by the Weimer 2005 and AMIE models, as well as on various empirical formulations,

globally- and hemispherically-integrated Joule heating rates were calculated during St Patrick's day storm of 2015. It is found that Joule heating rate estimates in the global circulation models, GITM and TIEGCM, for all external drivers, are generally higher in magnitude than any of the empirical models. Comparing $GITM_{AMIE}$ and $TIEGCM_{AMIE}$, it is found that they are in better agreement compared to $GITM_{Weimer}$ and $TIEGCM_{Weimer}$ in terms of amplitudes, peak-to-peak variability and inter-hemispheric asymmetry. In comparing the latter two models, it is found that Joule heating derived using

$TIEGCM_{Weimer}$ has lower amplitudes and also a lower peak-to-peak variability than $GITM_{Weimer}$. Significant variations that are observed are most likely attributed to the different precipitation models that are employed. This comparison is essential, as it reflects the default parameterizations of the two GCMs. On the other hand, Joule heating results derived using TIEGCM and GITM driven with AMIE data can be directly compared. Through a correlation analysis, and also by comparing the heating rates for a period of clear anti-correlation in the heating rate trend between the two models, it is found that both the

$TIEGCM_{AMIE}$ and $GITM_{AMIE}$ runs are strongly driven by the $SME$ index; this is not the case for the two Weimer runs, where $GITM_{Weimer}$ is strongly driven by the $SME$ index, which is not present in $TIEGCM_{Weimer}$. It is also found that Joule heating in $TIEGCM_{Weimer}$ is affected by the southward turnings of $B_z$ to a larger extent than $GITM_{Weimer}$. Joule heating calculated by the two GCMs using AMIE inputs shows similar values and peak-to-peak variability, compared to Joule heating derived using GCMs Weimer runs.

By integrating the Joule heating estimates separately in each hemisphere, it is found that $GITM_{Weimer}$ shows a larger degree of asymmetry during the main phase of the storm than $TIEGCM_{Weimer}$. Furthermore, by integrating Joule heating in time it is found that the cumulative Joule heating input to the thermosphere is larger as calculated in $GITM_{Weimer}$ in the SH and NH, followed by the various GCM runs, and then by the various empirical models. A factor of 2 difference is observed between the largest and smallest cumulative Joule heating when comparing different GCM runs, whereas a factor

of 25 is observed between the largest and smallest cumulative Joule heating, when all models (GCMs and empirical) are inter-compared. The localization (latitudinal and longitudinal distribution) of Joule heating in the two models exhibits slight variations, as depicted in characteristic time steps, as presented in Figure 4.

  In conclusion, as also demonstrated by the discrepancies in the above cross-comparisons between physics-based and empirical models, Joule heating remains to this date a quantity with many discrepancies in its estimation, showing large gaps in its

understanding and parameterization. At the same time, it is a quantity of great significance in LTI processes, as it determines to a great extent the overall energy budget, in particular during active solar and geomagnetic conditions. Thus, characterizing its magnitude, time evolution and variability within the latitude and altitude region where it maximizes and accurately



parameterizing Joule heating by solar and geomagnetic conditions are critical missing pieces in accurately understanding and modeling LTI processes. This demonstrates the currently limited knowledge about Joule heating and emphasizes the need for

comprehensive measurements, such as outlined in Sarris et al. (2023a), to accurately quantify Joule heating.

*Data availability.* Software used for calculation of Joule Heating is preserved at https://zenodo.org/records/10869507

*Author contributions.* ST and DB performed the TIEGCM runs. ST performed the calculations based on the empirical models. PP performed the GITM runs. ST, DB, PP worked on the analysis of the results and the preparation of the manuscript. TS, AR and GL contributed in the discussion of the results.

*Competing interests.* The authors declare that they have no conflict of interest.



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
