# Peer review of "Globally- and Hemispherically-Integrated Joule heating rates during the 17 March 2015 geomagnetic storm, according to Physics-based and Empirical Models"

_EGUsphere, 2025_

## Author Comment (AC2)

*We thank the reviewer for a very thorough evaluation of the manuscript, and for a number of very constructive comments. In the revised manuscript we have responded to all comments, which led to a significantly improved paper. In the following, our responses to the reviewers' comments are marked in blue italics, whereas the additions to the manuscript are marked in blue.*
* * *
**Reviewer #1 Evaluations:**
* * *
The paper *Globally- and Hemispherically-Integrated Joule heating rates during the 17 March 2015 geomagnetic storm, according to Physics-based and Empirical Models"* compares Joule heating (JH) estimates by the GITM and TIEGCM models, as well as by several empirical relations, for the storm of March 2015. While JH has various impacts, both fundamental and practical, its quantitative assessment is far from being settled. Numerical exercises like the one in the paper are useful to understand better the various models and eventually how to reach convergence. I shall be happy to recommend publication, once the issues listed below are clarified.

*We thank the reviewer for the positive comments on the usefulness of this study towards reaching convergence on this topic. We have taken into account the reviewer's comments, as described below.*
* * *
**1.** A major problem with results like those in the paper is the missing 'ground truth'. How to decide what result is better, which model to regard as more trustful? The problem is less critical when the various estimates are more or less similar, but this is hardly the case during storms - which are also of highest importance. Obviously, this major problem cannot be solved, but perhaps the authors can elaborate a bit, including hints to one or another more trustful JH proxy, if / when the case.

*We completely agree with the reviewer, that having a representative "ground truth" is indeed a key problem in the LTI, in particular during storm times, as the reviewer rightly points out. We also agree that this problem cannot be solved, except by new measurements, which has been highlighted in the revised manuscript. In response to the above, the following have been added in the paper:*

**Lines 355-367:** "It is noted that the comparative analyses presented above in Figures 1, 2, 3 and 4, and also the analyses in the works by Suji and Prince (2018), Raeder et al. (2001), Lu et al. (2023), Verkhoglyadova et al. (2017) and Zhu and Ridley (2016) can not, on their own, bring closure as to which model provides the most accurate estimate of the storm-time Joule heating. Instead, these analyses can provide the range of variability of Joule heating according to the different models and drivers, from which the uncertainty in estimating and predicting the state of the LTI can be approximated. Furthermore, the large discrepancies that are demonstrated by these model runs indicate that the exact quantification of Joule heating is a critically missing parameter in the LTI energetics. This is due to the lack of direct, comprehensive measurements in this region of the Earth geospace environment, as the altitude range where Joule heating

maximizes is too high to be sampled with probes on aerial vehicles and balloons and too low for typical spacecraft, making direct measurements therein challenging (see, e.g., Sarris (2019); Palmroth et al. (2021). Thus, the results of this study reinforce the current motion in the scientific community that efforts must be taken to close this gap, by dedicated missions that can provide all missing parameters required to estimate Joule heating, as outlined in equations 6, 8, and 15. Such mission concepts have been proposed and are being actively investigated (see, e.g., Sarris et al. (2020); Pfaff et al. (2022); ESA/NASA-ENLoTIS-Report (2024)."
* * *
**2.** The Weimer driving of GITM and TIEGCM emphasizes the importance of the precipitation model, as aptly discussed and concluded (L359-360, 417, 430-431). This is not surprising, since the precipitation model drives the (height-integrated) conductivity, which is an essential contributor to JH. It would be nice to comment a bit on the matter, including, if possible, a quantitative perspective (to what extent the differences in the results are indeed correlated to differences in conductivity).

*We agree with the insightful comment of the reviewer, in response to which the following comment has been added in the discussions section:*

**Lines 396-402:** "These results highlight the significance of particle precipitation in LTI energetics and the overall electrodynamic coupling within the LTI (e.g., Palmroth et al., 2021), since precipitation leads to increased ionospheric conductivity (Aksnes et al., 2004), which is an essential contributor to Joule heating. A further parametric study based on model runs under different parameterizations of particle precipitation would enable a thorough evaluation on the inter-relationship between particle precipitation, conductivity and Joule heating, and a quantitative assessment of the extent to which the differences in the model run results are indeed correlated to differences in particle precipitation and conductivity."
* * *
**3.** The differences between the AMIE results of GITM and TIEGCM are much smaller, but occasionally they are significant, like around 22 UT on March 17 (Figure 1 g). What could be the reason? As detailed in Section 2.3, the two codes use similar formulas for the Joule heating, the electric field is the same, and the precipitation, therefore the conductivity, is the same. Is the difference because of different boundary conditions? Different initializations? Different spatial / temporal resolutions, perhaps including different ways of addressing sub-grid? Some mix? Something else?

*Indeed, the differences between the AMIE results of GITM and TIEGCM are in general small, but there are, at times, significant differences. We have added a relevant paragraph discussing this, including potential reasons leading to these differences. We note that the exact cause of these differences could not be positively identified based on these simulation runs, and that it requires further investigation through a parametric study. The differences between the two models that could be the underlying cause have been listed in the discussions section, as follows:*

**Lines 333-341:** "A better agreement between the two GCMs is found when the AMIE model is used, even though at times there are significant differences even in this case, such as around 22 UT on March 17. As detailed in Section 2.3, the two codes use similar formulations for the calculation of Joule heating; also the electric field and the particle precipitation models are the same. Thus, a possible reason for these differences is related to the different spatial resolutions that are employed in the two models. Another possible reason is the different way that the two models treat vertical momentum and eddy diffusion: As discussed above, in GITM the complete vertical momentum equation is solved (see, e.g., Ullrich et al., 2010), which could lead to significant differences, in particular in the lower regions of the model, where also the eddy diffusion is expected to be larger. A more detailed parametric study could shed more light onto the causes of these discrepancies."
* * *
**4.** Any hint on why the GCMs provide systematically higher JH as compared to the empirical formulas? (e.g., L369, 427)

*The following possible explanation has been added in the Discussions section of the revised manuscript:*

**Lines 406-409:** "A possible reason is that empirical models in general rely on historical data and statistical models that may not adequately capture the complex, non-linear dynamics and feedbacks that drive extreme events, such as solar storms. GCMs on the other hand, while still having limitations, attempt to simulate these complex processes, potentially offering a more realistic representation of extreme events."
* * *
**5.** I do not fully understand panels f and g of Figure 1. The hemispheric powers in panel f, labeled hpower and expressed in GW, do not add up to the global power in panel g, labelled Joule heating and expressed in GWatts (why not GW?). Probably I am missing something. Please clarify, in the caption of the figure and in the para at L283-291. Further on, panels a and b of Figure 2, seem to (rightly) add up to panel g of Figure 1.

*We thank the reviewer for spotting this. Indeed, the correct values of the hemispheric power from the various models are included in Figure 2, and are different from the values in Figure 1 panel f. The results in panel f of Figure 1 are left from a previous iteration of the paper. In the revised manuscript we have removed this panel altogether. Also GWatts has been changed to GW in the revised manuscript.*
* * *
**6.** Others

**6.1** L10-11: With AMIE, TIEGCM and GITM are rather similar, with TIEGCM occasionally higher (see also point 3).

*We thank the reviewer for spotting this error in the abstract; we have rephrased the abstract as follows:*

**Lines 11-13:** "TIE-GCM provides lower estimates of the heating rates compared to GITM when the Weimer 2005 model is used as driver, whereas TIE-GCM and GITM give rather similar estimates when the AMIE model is used, with TIE-GCM occasionally giving higher estimates."
* * *
**6.2** L66: I think AMIE has a considerably broader scope, not just to mitigate the discrepancies between JH estimates.

*Indeed, this wording is not accurate; we have rephrased this as follows:*

**Lines 66-67:** "Various data assimilation methods have been developed which can be used to mitigate the discrepancy"
* * *
**6.3** L128: Please explain briefly 'Photoelectron heating is based on a streamlined connection'.

*Indeed, we agree with the reviewer that this description is not straight forward; this has been replaced as follows:*

**Lines 130-131:** "Photoelectron heating is calculated using an empirical model that simplifies the complex process of photoelectron production and energy deposition in the ionosphere."
* * *
**6.4** L137: Please describe TIEGCM briefly, similar to GITM one line above.

*The following description of the calculation of Joule heating in TIE-GCM has been added:*

**Lines 140-142:** "Joule heating in TIE-GCM is obtained via the calculation of the Pedersen conductivity and the electric field in the reference frame of the neutral wind, following the approach by Lu et al. (1995)."
* * *
**6.5** L138: 'the equivalence of the two methodologies is derived,' => Please re-phrase. What is derived / demonstrated is the equivalence of the formulas used to compute JH. The methodologies are not really equivalent, as shown by the different results.

*Indeed, the equivalence of the formulas is what is derived, whereas the methodologies are not equivalent; this has been rephrased as follows:*

**Line 142:** "the equivalence of the formulas used to calculate Joule heating in the two models is derived, highlighting the assumptions used in each methodology"
* * *
**6.6** L141: 'by assuming a quasi-steady state' => Does this fit with a storm event?

*This is a very valid point, and the effects and implications of this assumption need to be investigated; we note, however, that this the standard approach that is followed in the estimations done in TIE-GCM. The following text and references to papers describing TIE-GCM have been added:*

**Line 148-152:** "It is understood that the assumption of quasi-steady state does not necessarily fit with a storm-time event, however this is a common assumption that is followed in TIE-GCM simulations (see, e.g., Lu et al., 1995; Qian et al., 2014; Richmond and Maute, 2014). The effects and implications of this assumption need to be investigated, but such investigation is beyond the scope of this study."
* * *
**6.7** L308: 'the time of maximum percentage difference' => Is this indicated by dotted line C?

*We thank the reviewer for pointing this out. Actually, this time is not indicated by a line in Figures 1, 2 and 3; this has been clarified in the text as follows:*

**Lines 316-321:** Three characteristic times during St Patrick's day storm on 17 March 2015 are plotted: 06:20 UT (left-hand side panels), marked as line A in **Figures 1**, **2** and **3**, corresponding to the beginning of the storm;  14:10 UT (middle panels), corresponding to the time of maximum percentage difference in Joule heating between GITM$_{Weimer}$ and TIEGCM$_{Weimer}$ (not marked with a line in the above figures) and 22:50 UT (right-hand side panels), marked as line D in the above figures, which corresponds to the peak of the storm, as indicated by the minimum in Dst.
* * *
**6.8** L314 (and 446-447): 'shows notable similarities' => Except for the middle plots of a) and b).

*The discussion  on the similarities has been updated as follows:*

**Lines 322-326:** The comparisons between **Figure 4** panels **(a)** and **(b)** at the top and panels **(c)** and **(d)** show that, apart for the large differences in the amplitudes of Joule heating between GITM and TIEGCM, the distribution of Joule heating in longitude and latitude shows notable similarities, except for the middle plots of (a) and (b), which represent calculation using the Weimer model, with slight variations in the localization and extent of the spatial structures where Joule heating appears.
* * *
**6.9** Caption of Figure 1: Include explanation of dotted lines A, B, C, D (e.g., see text?)

*The following sentence has been added in the caption of Fig. 1:*

**Figure 1 caption:** "The vertical dashed lines (A, B, C, D) mark specific time steps and are discussed in the text."
* * *
**6.10** L340-341, 367-368, 435: Please re-phrase (replace 'driven by' with 'related to'?). I

understand that SME is one of the parameters that drive the simulations, but this does not mean that SME / the electrojet drives JH (in particular, the electrojet is typically dominated by Hall current). Both JH and SME are (mostly) driven by magnetospheric dynamics via M-I-T coupling.

*We completely agree with the reviewer; 'driven by' has been replaced with 'related to' in the instances mentioned above, and the related sentence has been corrected as follows:*

**Line 371:** "Through this correlation analysis, it is found that Joule heating is strongly related to the SME electrojet strength"
* * *
**7.** Typos and alike

L19, 22, 25, 29, 30, 39, 41, 42, 56-57, 76, 112, 115, 244, 325, 338, 401: References indicated by \citet instead of \citep. L23: Space-X => Starlink (?); L32: found in => related to (?); L34: Delete the bracket after 2023a; L34-35: Move 'due to the large drag' at the end of the sentence (it only affects the satellites, not the balloons); L35: Delete 'current' (?); L67: Explain LDFF; L78 and 600-601: Duplicate of 598-599; L97: makeS; L99: Eddy => eddy; L118: comma before and after j; L137: outlined in => by; L143: Move 'component of the electric field' before 'parallel' one line above; L151: Ohm's LAW; L152: among => along; L181: based ON; Eq. 17: '-1' should be aligned with 'k' => $z_{k-1}$; L185: analysis => techniques? calculations?; L245: (g) => (e); L253: n/cc => $cm^{-3}$; L257: are => is; L275 and 281: Resolve the question marks; L281: thRough; L305: is shown as a polar plots => are shown as polar plots; L310: Figure 4 => Figure 4 c, d; L312: (c) => (b) at, (b) => (c); Caption of Figure 1, third line: (d) => (f); L330: addresses => addressed.

*We thank the reviewer for a very thorough reading of the manuscript and for spotting these; all typos have been corrected and are marked with blue fonts in the revised manuscript.*

---

## Author Response (AR2)

We thank the reviewer for these comments. We have added a description of the vertical lines in the figure caption of Figure 1. We also thank the reviewer for spotting typos, which have been corrected in the revised manuscript.